# OpenReview forum: "DP-Fusion: Token-Level Differentially Private Inference for Large Language Models"
_ICLR.cc/2026/Conference — ICLR 2026 Poster_

### Official Review · Reviewer_jYDA · 2025-10-24

**Soundness:** 3
**Presentation:** 3
**Contribution:** 3
**Rating:** 6
**Confidence:** 4

**Summary:**

This paper introduces DP-Fusion, which bounds the privacy leakage of a document by mixing public and private distributions with respect to pre-defined sensitive tokens. It begins by classifying sensitive/non-sensitive tokens and running parallel generation. The logit distributions are then blended so that the output preserves high quality while the privacy of sensitive tokens are also guaranteed. It is claimed to yield better privacy-utility trade-off than NER-based anonymization and prior DP inference methods. Extensive experiments exhibit the effectiveness of the proposed work.

**Strengths:**

- The paper is well-written and Figure 2 clearly depicts the pipeline of the proposed method.
- DP prediction for document/prompt santization is an important problem.
- The idea of mixing public and private distributions, followed by a group-wise privacy accounting according to the entity type is intuitive and reasonable.

**Weaknesses:**

- I find some similar works on combining DP and mixing logit distribution. Could the author clarify the major difference against these works?
- I have one performance concern regarding the comparison against the No DPI - NER baseline. According to Figure 3, DP-Fusion have ~50% win-rate against No DPI - NER. Beisdes, as indicated by Table 1, the utility and privacy of these two methods are also similar.
- The PII tagger seems to still have about 4% FN, leaving potential threat to practical highly-sensitive phrases.

[1] Private prediction for large-scale synthetic text generation. https://arxiv.org/pdf/2407.12108
[2] Differentially Private Next-Token Prediction of Large Language Models. https://arxiv.org/abs/2403.15638

**Questions:**

- What if the No DPI - NER replaces the sensitive tokens with entity type (which should be non-sensitive)? For example, replace a person_name to [applicant]. Will this variant achieves a better utility?
- Could the authors discuss some workarounds when PII tagger has some FN cases?

---

> ### Author Response · Authors · 2025-11-20
> **We thank the reviewer for their valuable feedback.**
>
> We appreciate the reviewer's positive comments on our write-up, motivation, and approach. Please find our response below regarding alternative methods and comparisons with baselines.
>
> > I find some similar works ...
>
> We thank the reviewer for highlighting these works. PMixED [2] is an important foundation for our approach. However, PMixED operates at training time, whereas we study the LLM inference setting under a fundamentally different threat model and privacy game.
>
> The other highlighted work considers a different setting, where the goal is to generate synthetic text similar to a supplied text distribution, typically for training language models with DP (i.e., the task “Generate text similar to:”). In that setup, the public distribution is simply a generic non-sensitive prompt (e.g., an instruction or template).
>
> However, it may be possible to use their sampling approach as an alternative way to mix distributions across privacy groups, and we will explore this in future work. Although, the core ideas of clipping and noise addition are already represented in our baselines, DP-Prompt [3] and DP-Decoding [4], respectively.
>
> We would also like to refer the reviewer to our response to a similar point (“How exactly is DP-Fusion different from the mentioned methods…”) raised by reviewer 7yFi. We will update Section 4 to explain background in more detail.
>
> > I have one performance concern ...
>
> We thank the reviewer for this observation. We would also like to refer them to our response to reviewer 7yFi, who raised a similar point in Weakness 6.
>
> In Figure 3, DP-Fusion performs better than No DPI–NER at ($\alpha\beta$ = 0.1), with a 56% win rate, although the gap is not large. The gap between is expected to grow with the amount of PII in the dataset, since DP-Fusion can preserve residual utility from private tokens. We were limited by available datasets, but we ran additional experiments using a single privacy group ($m=1$) and on a different dataset, which we detail below.
>
> The single-group implementation of DP-Fusion is described in Appendix A.19. We also include an alternative utility metric following prior work [6], using cosine similarity between text embeddings from *sentence-transformers/all-MiniLM-L6-v2*. We compute similarity between each generated paraphrase and the original document.
>
> At $\alpha\beta_{i}=0.01$, DP-Fusion achieves higher utility than No DPI–NER (0.816 vs. 0.8093 cosine similarity) while also providing lower LOSS ASR (0.263 vs. 0.277).
>
> On the medical MACCROBAT dataset (Appendix A.20), we observe a much larger gap. DP-Fusion paraphrases exhibit significantly higher similarity to the original document (0.6348 vs. 0.4972) while maintaining comparable LOSS ASR (0.125 vs. 0.117). We posit that in this dataset, private tokens contribute substantially to semantic content, reflected in the large gap between No-DPI NER (0.4972) and No-DPI Original Document (0.8396), compared to the much smaller gap in ECHR (0.8092 vs. 0.8254). This allows DP-Fusion to retain more task-relevant information and thus achieve higher utility.
>
>
> We will address this limitation of the experiments with the ECHR dataset, where the PII appears to have limited impact on output quality, explicitly in the revised paper, while referring to the additional experiments in Appendix A.19 and A.20, where the gap over No DPI–NER widens. Our method is rigorous, and as we relax $\epsilon$ we consistently observe improved quality. We thank the reviewer for raising this important point.

---

> > ### Author Response · Authors · 2025-11-20
> > **Response to other weaknesses, questions and references.**
> >
> > > The PII tagger seems to still have about 4% FN ...
> >
> >
> > We acknowledge that unintended leakage may occur if the tagger is inaccurate. Our theoretical guarantees rely on correct identification of PII; otherwise, only empirical guarantees apply. Nonetheless, it is encouraging that PII taggers continue to improve, which directly strengthens the effectiveness of our method.
> >
> >
> > It is also important to note that the PII tagger itself can be tuned (e.g., by lowering the detection threshold in a BERT-based NER model) to aggressively reduce false negatives at the expense of more false positives. Any residual false negatives are further safeguarded by the paraphrasing step, which has been shown to capture and transform many of these cases [3,5].
> >
> > Developing accurate PII taggers is orthogonal to our work and DP-Fusion directly benefits from improvements in tagging systems. We cover this point in Section 6 on the "Role of Tagging", and we will update that section to more clearly explain the issues points raised.
> >
> >
> >
> > > What if the No DPI - NER replaces the sensitive tokens with entity type ...
> >
> > We thank the reviewer for this suggestion. Using semantically meaningful placeholders could indeed increase utility, since downstream models would retain some information about the type of the redacted token. However, this approach can also leak partial information about the underlying PII and therefore weakens the intended privacy guarantees.
> >
> > In our current implementation, we intentionally avoid this leakage and instead enforce that all document variants, both the public version $X_{\text{pub}}$ and the selectively revealed versions $X_{\text{pub}} \cup X_i$ (Section 4.1), contain exactly the same number of tokens. This design removes token-length artifacts, ensuring that any change in the model’s output distribution is attributable solely to the reintroduced private content, rather than to differences in length or tokenization.
> >
> > To achieve this, we use the placeholder "\_", which is treated as a single subword token in most tokenizers, and *we insert exactly as many placeholders as the number of private tokens removed* i.e we insert multiple "\_" to match the token length of the PII.
> >
> > For example:
> > Consider the date “21st Nov 2025”, which the tokenizer splits into six tokens:
> >
> > * `21`
> > * `st`
> > * `Nov`
> > * ` `
> > * `202`
> > * `5`
> >
> > Since the date corresponds to six tokens, we replace it with six underscore placeholders:
> > ```
> > ______
> > ```
> > This guarantees that all document variants $X_{\text{pub}}$ or $X_{\text{pub}} \cup X_i$ have the same number of tokens.
> >
> > However, we agree that alternative placeholder strategies are an interesting direction for improving utility. We do not evaluate them in this work and leave such strategies for future. We will revise Section 4.1 to explain our current placeholder approach clearly.
> >
> > > Could the authors discuss some workarounds when PII tagger has some FN cases
> >
> > Typical taggers such as BERT-based models output confidence scores, allowing thresholds to be raised to reduce false negatives at the cost of higher false positives. Additionally, as noted earlier, prompt-engineered LLMs provide more robust PII tagging.
> >
> > *We value your feedback and thank you for it. We hope our response addresses all of your concerns.*
> >
> >
> > ## References
> >
> > [1] Amin et al. Private prediction for large-scale synthetic text generation. EMNLP, 2024.
> >
> > [2] Flemings et al. Differentially Private Next-Token Prediction of Large Language Models. NAACL, 2024.
> >
> > [3] Utpala et al. Locally Differentially Private Document Generation using Zero‑Shot Prompting. EMNLP, 2023.
> >
> > [4] Majmudar et al. Differentially Private Decoding in Large Language Models. NAACL, 2022.
> >
> > [5] Staab et al. Large Language Models Are Advanced Anonymizers. ICLR, 2024.
> >
> > [6]  Lau et al. Understanding the Effects of Human‑written Paraphrases in LLM‑generated Text Detection. 2024.

---

> > > ### Comment · Reviewer_jYDA · 2025-11-27
> > >
> > > Thank the authors for the detailed response. Most of my concerns have been resolved, and i am happy to maintain the positive score.
> > >
> > > I hope the authors can elaborate on the above-mentioned limitations and clarifications, and include these discussions in the final version.

---

> > > > ### Author Response · Authors · 2025-11-28
> > > >
> > > > We appreciate the reviewer for taking the time to go through our rebuttal and for maintaining their positive score. We are currently working on incorporating all proposed updates and will upload the revised manuscript in the coming days.

---

### Official Review · Reviewer_Vgi2 · 2025-10-26

**Soundness:** 3
**Presentation:** 2
**Contribution:** 2
**Rating:** 4
**Confidence:** 3

**Summary:**

The paper proposes the DP-FUSION, a novel method for document privatization.  DP-FUSION enhances privacy during inference in large language models (LLMs) by bounding the influence of sensitive tokens, such as personally identifiable information (PII), on the output. DP-FUSION operates by redacting sensitive tokens, generating baseline and private distributions, mixing them with a controlled privacy/utility trade-off parameter ε, and sampling tokens to produce privatized documents. It offers provable differential privacy (DP) guarantees and mitigates risks like jailbreak attacks while preserving text quality. Experimental results suggest that DP-FUSION achieves much lower perplexity than related DPI methods.

**Strengths:**

1. Provable Privacy: DP-FUSION provides formal (ε, δ)-DP and Rényi DP guarantees, ensuring bounded leakage of sensitive information.


2. Improved Utility: DP-FUSION achieves significantly lower perplexity than prior DPI methods, balancing privacy and text quality.


3. Flexibility: DP-FUSION allows per-group privacy budgets, enabling tailored protection for different sensitive token types (e.g., names, dates).

**Weaknesses:**

1. My major concern is its Limited Scope. DP-FUSION only focuses on document privatization, potentially limiting applicability to other LLM use cases like real-time chat or tool-augmented inference. The proposed DP-FUSION pipeline seems to only consider the PII privacy and ignore the privacy that may be implicitly inferred from the document context.

2. The experiments of DP-FUSION only consider its paraphrase based on perplexity and the LLM judge. From my point of view, applying these paraphrased documents to downstream tasks and measuring the task utility could be a much better utility indicator.

3. Some citation typos in related works, such as Section 2.1 of the BACKGROUND. Using \citep would be better to improve the readability.

**Questions:**

1. What happens if the local tagger cannot identify the PII? Is there any evidence to support that NER-based taggers are reliable? There are a lot of tricks to bypass such filters, such as using hashtags.

2. What happens if the private information is implicitly hidden under the semantics of a document? For example, a document of multi-round conversations may encode more private information other than PII?

3. How can the paraphrased documents be used? Will these documents be helpful for real-world downstream tasks?

---

> ### Author Response · Authors · 2025-11-21
> **We thank the reviewer for their feedback.**
>
> We appreciate that they recognize DP-Fusion’s ability to provably bound leakage with flexible, per-group $\epsilon$, while achieving significantly better tradeoffs than prior DPI methods. Please find below our response to the reviewer’s concerns regarding scope and downstream utility.
>
> > ... DP-FUSION only focuses on document privatization ...
>
> We focus on document privatisation in this work as a benchmark to demonstrate DP-Fusion’s effectiveness as a DPI mechanism. This task is well studied and directly reflects how real-world systems implement privacy protection [1]. Our threat model is analogous to the threat model considered in existing prompt sanitization work such as Preempt [2] and InferDPT / RANTEXT [3], as well as prior local DP text anonymization approaches including CusText [4] and SanText [5].
>
> Our DPI mechanism applies also to real-time chat, tool-augmented workflows, and other LLM applications, but we did not evaluate these due to a lack of standardized datasets and benchmarks. All such settings ultimately reduce to constructing an input context $C_{input}$ of $M$ tokens and generating $N$ output tokens. A DPI mechanism can assign a privacy budget $\epsilon$ to any subset ($K \subseteq C_{input}$), given an oracle (typically an LLM) that identifies these tokens. DP-Fusion then ensures this budget is respected across the $N$ generated tokens by controlling parameters such as $\beta$.
>
> Beyond PII privacy, DP-Fusion can be used for security-related tasks, which we show in Appendix A.21 where we defend against jailbreak-style prompt-injection attacks.
>
> We reply to the reviewer's suggestion by demonstrating DP-Fusion in the context of a simple chatbot. For this purpose, we construct a custom, multiple-choice questionnaire from ECHR document chunks and evaluate various DPI mechanisms as methods for privatizing the context while generating in a question-answering setting.
>
> We sample 200-token (Qwen-2.5 tokenizer) chunks from each document. ***Please refer to our response to reviewer EZSD (Weakness 1) for the full statistics of this dataset.*** We define a question for each chunk by prompting GPT-4o, which also generates the correct option and distractors. We then pass the question, context, and options into an evaluation prompt. ***The questions and the full evaluation prompt are also provided in our response to reviewer EZSD.***
>
> To simulate real-world chat settings, we apply different DPI methods during output generation, treating them as mechanisms to prevent the private context from leaking through the produced answers.
>
> For instance, DP-Fusion, under the single-group implementation (Appendix A.19), assigns an $\epsilon$ to the private tokens in the context and then generates output tokens by sampling from the mixed distribution to produce an answer.
>
> We measure accuracy by extracting the option selected by the LLM (i.e., the token appearing after “The answer token is:”) and comparing it with the correct answer.
>
> The below table describes the accuracy with different methods. We also include the ASR for the strongest attack, LOSS attack (Section 4.3) from Table 1.
>
> | Method      | Parameters          | Accuracy        | LOSS   |
> | ----------- | ------------------- | --------------- | ------ |
> | DP-Decoding | λ=0.1             | 32%             | 15.67% |
> | DP-Decoding | λ=0.9             | 96%             | 66.00% |
> | DP-Prompt   | w=50, T=0.75 | 90%             | 56.67% |
> | DP-Prompt   | w=50, T=1.75 | 57%             | 28.67% |
> | DP-Prompt   | w=5, T=0.75  | 24%             | 26.67% |
> | DP-Prompt   | w=5, T=1.75  | 27%             | 17.33% |
> | DP-Fusion   | $\alpha\beta_{i}$=0.001           | 53%             | 28.00% |
> | DP-Fusion   | $\alpha\beta_{i}$=0.01            | 52%             | 26.30% |
> | DP-Fusion   | $\alpha\beta_{i}$=0.1             | 51%             | 29.30% |
> | DP-Fusion   | $\alpha\beta_{i}$=5.0             | 86%             | 45.70% |
> | DP-Fusion   | $\alpha\beta_{i}$=10.0            | 99%             | 56.30% |
> | No DPI      | NER                 | 47%             | 62.70% |
> | No DPI      | Original Document   | 100%            | 27.70% |
>
>
> DP-Fusion demonstrates stronger utility–privacy tradeoffs across most regimes. At comparable privacy levels (LOSS ASR $\approx$ 26%–30%), it attains 51–53% accuracy, whereas DP baselines range from 24–57%. At higher privacy levels (LOSS $\approx$ 45%–57%), DP-Fusion reaches 86–99% accuracy, while DP baselines achieve 90–96% but at strictly worse privacy. Only at very low LOSS (<20%) do baselines perform better. Conversely, at matched utility (50–53% or 86–99% accuracy), DP-Fusion consistently achieves lower LOSS ASR than the DP baselines, which only outperform it in the low-accuracy range (25–32%).
>
> We will include the full prompt used to construct this dataset, together with its description and dataset statistics, in the appendix. We will also add the multiple-choice evaluation prompt and he corresponding results.

---

> > ### Author Response · Authors · 2025-11-21
> > **Response to other weaknesses, questions and references.**
> >
> > > The experiments ... paraphrase based on perplexity ...
> >
> > Thank you for raising this suggestion. We would like to point out that we have validated quality via (i) perplexity (ii) win-rates via LLM-as-a-Judge and (iii) a qualitative analysis. To further assess downstream performance on paraphrased documents, we conduct an additional experiment in which we apply various DP methods and baselines to generate a privatized version of each chunk, and then evaluate these outputs using the same chat template described previously.
> >
> > The below table describes the accuracy with different methods as the privatising mechanisms. We also include the ASR for the strongest attack, LOSS attack (Section 4.3) from Table 1.
> >
> > | Method      | Parameters               | Accuracy | ASR % - LOSS   |
> > | ----------- | ------------------------ | -------- | ------ |
> > | No DPI      | Original Document        | 98%      | 62.70% |
> > | No DPI      | NER                      | 34%      | 27.70% |
> > | DP-Decoding | λ=0.1                  | 23%      | 15.67% |
> > | DP-Decoding | λ=0.9                  | 70%      | 66.00% |
> > | DP-Prompt   | w=5, T=0.75          | 31%      | 26.67% |
> > | DP-Prompt   | w=5, T=1.75          | 32%      | 17.33% |
> > | DP-Prompt   | w=50, T=0.75         | 90%      | 56.67% |
> > | DP-Prompt   | w=50, T=1.75         | 33%      | 28.67% |
> > | DP-Fusion   | $\alpha\beta_{i}$=0.001| 36%      | 28.00% |
> > | DP-Fusion   | $\alpha\beta_{i}$=0.01 | 37%      | 26.30% |
> > | DP-Fusion   | $\alpha\beta_{i}$=0.1  | 38%      | 29.30% |
> > | DP-Fusion   | $\alpha\beta_{i}$=5.0  | 60%      | 45.70% |
> > | DP-Fusion   | $\alpha\beta_{i}$=10.0 | 85%      | 56.30% |
> >
> > For DP-Fusion, we use the faster single-group setting described in Appendix A.19. At the same privacy range (ASR $\approx$ 0.26–0.29), DP-Fusion achieves the highest utility (38% accuracy). At ($\alpha\beta_i$ = 0.01), DP-Fusion offers better trade-offs than No DPI–NER, achieving higher utility (34% vs. 37%) while providing more privacy (lower ASR 27.7% vs. 26.3%). As ($\alpha\beta_i$) increases, the trade-offs move smoothly from being closer to the No DPI–NER setting toward the No DPI–Original Document setting.
> >
> > > Some citation typos ...
> >
> > We thank the reviewer for the suggestion and have revised the paper accordingly.
> >
> > > What happens ... cannot identify the PII ...
> >
> > This is a great question. We acknowledge that relying on a fixed PII tagging model can have errors and missed PII will **not** be covered under the theoretical privacy guarantees. For the theoretical guarantees to hold, we rely on expert annotators who carefully tag *all* PII (recall of 100%), even if the precision is low. Interestingly, the gap between No-DPI NER and our method increases as the precision decreases, as more tokens are labelled as sensitive but only our method, DP-Fusion, allows retaining some utility from sensitive tokens. In our paper we use ground-truth labels with both perfect recall and precision (i.e., 100\%). We would like to point out that in practice scoring-based NER systems allow trading off lower precision in exchange for higher recall.
> >
> > On TAB-ECHR, taggers already achieve low false negative (FN) rates (3.9% FN, 85.4% F1; Appendix A.17). DP-Fusion’s performance with a commonly used tagger in practice is shown in Appendix A.18.
> >
> > We cover this point in Section 6 on the **Role of Tagging**, and we will update that section to more clearly explain the issues points raised.
> >
> >
> > > What happens if the private information is implicitly hidden ...
> >
> > The multi-round conversation setting is interesting and could be a topic for future work. We focus on the single-round setting and acknowledge that there could be documents where private tokens are more challenging to identify. For any missed token, our method offers only empirical privacy protection. Developing accurate PII tagging oracles is an important, but orthogonal direction to our work. We would like to highlight that DP-Fusion directly benefits from improvements in tagging systems which appear to become increasingly capable at accurately tagging sensitive tokens.
> >
> > > How ... documents be used ...
> >
> > We appreciate the reviewer’s concern regarding downstream utility for end users of DP-Fusion. *We hope our experiments address this concern and illustrate our method's usefulness for downstream tasks such as question answering in a chatbot setting.*
> >
> > ## References
> >
> > [1] Utpala et al. Locally Differentially Private Document Generation using Zero‑Shot Prompting. EMNLP, 2023.
> >
> > [2] Roy Chowdhury et al. Prϵϵmpt: Sanitizing Sensitive Prompts for LLMs. 2025.
> >
> > [3] Tong et al. InferDPT: Privacy‑Preserving Inference for Black‑box Large Language Models. 2023.
> >
> > [4] Chen et al. CusText: A Customized Text Sanitization Mechanism with Differential Privacy. ACL 2022.
> >
> > [5] Yue et al. SanText: Differentially Private and Semantically Coherent Text Sanitization. ACL, 2021.
> >
> > [6] Staab et al. Large Language Models Are Advanced Anonymizers. ICLR, 2024.

---

> > > ### Comment · Reviewer_Vgi2 · 2025-11-22
> > > **Rebuttal Acknowledgement**
> > >
> > > Dear Authors,
> > >
> > > Thanks for your efforts and additional experiments to clarify a few of my misunderstandings. Still, I am not quite convinced by the paper's claims and limitations on PII.
> > >
> > > 1. "Our DPI mechanism applies also to real-time chat, tool-augmented workflows, and other LLM applications, but we did not evaluate these due to a lack of standardized datasets and benchmarks."
> > >
> > > There are many benchmarks regarding multi-turn chats, tool usage, and other complicated applications. For example, you may consider such as BFCL and Toucan [1, 2]. If you claim that DP-Fusion works on these complicated use cases, experiments on these benchmarks are definitely necessary to support your claim.
> > >
> > > 2. The privacy scenarios beyond PII are not captured by the evaluation. Given that there is a series of ongoing works on inference attacks, the paper fails to address this concern.
> > >
> > > Therefore, I decide to keep my score.
> > >
> > >
> > > [1] Shishir G Patil and Huanzhi Mao and Fanjia Yan and Charlie Cheng-Jie Ji and Vishnu Suresh and Ion Stoica and Joseph E. Gonzalez, The Berkeley Function Calling Leaderboard ({BFCL}): From Tool Use to Agentic Evaluation of Large Language Models, ICML 25.
> > >
> > > [2] Xu, Zhangchen, et al. "Toucan: Synthesizing 1.5 m tool-agentic data from real-world mcp environments." arXiv preprint arXiv:2510.01179 (2025).

---

> > > > ### Author Response · Authors · 2025-11-24
> > > > **We thank the reviewer for taking the time to go through our rebuttal.**
> > > >
> > > > >  If you claim that DP-Fusion works on these complicated use cases, experiments on these benchmarks are definitely necessary to support your claim.
> > > >
> > > >
> > > > We begin by summarising the experiments conducted so far to evaluate the applicability of DPI mechanisms:
> > > >
> > > > 1. Paraphrasing (Perplexity): Section 5.2 (lines 315-319).
> > > > 2. Paraphrasing (LLM-as-a-judge): Section 5.3 (lines 324-327).
> > > > 3. Paraphrasing (Cosine similarity): Appendix A.19 (lines 1178-1181), A.20 (lines 1240-1241).
> > > > 4. Jailbreaking defense evaluation: Appendix A.21 (Table 13).
> > > > 5. Downstream task - MCQ accuracy: Included in the rebuttal (response to weakness 2).
> > > > 6. Downstream task - MCQ in a live chat-style setting: Included in the rebuttal (response to weakness 1).
> > > >
> > > > Through these experiments, we have attempted to cover the full spectrum of how DPI mechanisms would be used in real-world settings, under a threat model where private tokens (PII or any user-defined sensitive tokens) appear in the LLM context.
> > > >
> > > > We agree that agentic tool use is an important emerging LLM application. At the same time, these systems ultimately operate by repeatedly invoking the LLM itself (typically via `llm.generate`) to produce text or structured outputs specifying which tool to call next. The tools are executed by the application, not the model, so the only component that actually needs to be privatized is the LLM inference step. Therefore, applying our method simply requires replacing standard inference (`llm.generate`) with our private inference mechanism, together with labels for the private tokens in the input.
> > > >
> > > > We note that the reviewer’s remaining concerns relate not to correctness or novelty, but to seeing additional results on tool-use benchmarks. We did not run such evaluations because the privacy threat model for this setting is not clearly defined. We would greatly appreciate clarification from the reviewer on what exact threat model they have in mind for tool-augmented workflows. Specifically, where should private tokens be labelled in such a setup? Are these private tokens intended to appear in the user intent, the tool arguments, or elsewhere?
> > > >
> > > > When we noted the “*lack of standardized datasets and benchmarks*,” we were referring precisely to this point: real-time chat or agentic datasets where private tokens are explicitly present and marked in the context of all LLM calls are limited.
> > > >
> > > > > The privacy scenarios beyond PII are not captured by the evaluation.
> > > >
> > > > We would like to point out that we already evaluate two probabilistic membership-inference attacks, MIN-K and LOSS, both of which achieve high ASR. Given their white-box nature, these attacks are expected to be strictly stronger than simple inferential attackers (e.g., prompting). At the same time, we would be happy to take a closer look if the reviewer could clarify which additional inferential attackers they are referring to in this context.
> > > >
> > > > We thank the reviewer again for the constructive discussion around more complex use-cases. We would appreciate any clarification they can provide on our questions above.

---

> > > > > ### Author Response · Authors · 2025-11-28
> > > > >
> > > > > We look forward to the reviewer’s response, especially regarding the privacy threat model and setup for the proposed tool-calling benchmark.

---

### Official Review · Reviewer_7yFi · 2025-10-30

**Soundness:** 3
**Presentation:** 3
**Contribution:** 3
**Rating:** 8
**Confidence:** 4

**Summary:**

The authors propose a new method called DP-Fusion that aims to protect sensitive data in the LLM's context while at the same time maintaining high generation / response quality. The authors provide provable bounds on the influence of certain tagged tokens in the input on the output distribution. In addition, the authors provide empirical experiments comparing their method to several baselines based on perplexity evaluated on the original document as well as using LLM-as-a-judge-based win rates on the output responses. Overall, the authors find their method to outperform most baselines while maintaing high output quality and tunable levels of privacy.

**Strengths:**

Overall, the paper is well written. In particular, I liked the authors had a full section dedicated to describing the threat model. In addition, Figure 2 is quite clear in terms of providing an overview of the proposed method. The results also generally seem quite strong (except maybe with respect to No DPI - NER), providing much better perplexity values while also having relatively low ASR values for the attacks.

**Weaknesses:**

1. Page 4: The first paragraph of Section 4 could be strengthened: How exactly is DP Fusion different from the mentioned methods like PMixED, PATE and SUBMIX? Right now the differences are not very clear.
2. Page 6: In line 3 of the algorithm box, it seems D’ which contains D and which itself is defined as the union over the public AND the private tokens is passed to the LLM to compute the public distribution. However, doesn’t passing D’ mean that the LLM has access to the private tokens as well? So then why is the resulting distribution a public one?
3. Page 6: Both min-k and LOSS attack could use maybe one more sentence of explanation: how does min-k actually infer the secrets once the min k tokens have been identified? And what does the loss attack do with the loss for each of the candidate secrets? As in, how does it use those to predict the true secret?
4. Page 7: If I understand correctly, it seems only one model is used (Qwen 2.5 7B Instruct) for all the experiments. the paper could be strengthened by including experiments on a few more models, ideally from different families (e.g. one Llama and one Phi model or so).
5. Table 1: It would be good to add standard errors to get a better sense of statistical significance.
6. Table 1: It seems somewhat disappointing that after all this DP machinery, the proposed method does roughly the same as just No DPI - NER in terms of perplexity and ASR. Could the authors comment on this?

**Questions:**

1. Page 2: “our experiments show that attackers can still reliably infer sensitive information when they know which model was used to create the paraphrased text” → Where are the experiments for this in the paper?
2. Page 3: “we show that inferential white-box attackers can infer membership at a high success rate without jailbreaking” → Where are the experiments for this in the paper? I thought the attacker was mentioned to be gray-box in the threat model?
3. Page 5: What’s the importance of looking at symmetric Renyi divergence? It seems in Theorem 1 nothing was mentioned with respect to symmetry?
4. Page 5: “\beta is the main controller of privacy-vs-utlity and a proxy for \epsilon in our DPI mechanism” → Could the authors clarify exactly how \beta is a proxy for \epsilon?
5. Page 5: Why is \alpha set to 2?
6. Page 6: “Section 4.2 gives an upper bound on the attacker’s advantage measured in the recovery game”. However, I couldn’t find any explicit upper bound on the advantage as defined in equation 4 - could the authors clarify this?
7. Page 7: “we modify the base prompt with instructions like ‘produce a natural paraphrase of this for ensuring privacy’”. However, I couldn’t find this example in the prompts in the appendix. So does this mean “No DPI - Original Document” is just a raw paraphrase by the LLM, or the LLM is specifically asked to ensure privacy?
8. Page 7: “Although theoretical guarantees are not directly comparable across methods, plotting utility versus the reported ϵ still illustrates the trade-off each method achieves.” → So does this mean the epsilons of different methods are comparable or no? And if they are, then what exactly is not comparable across methods?
9. Page 8: “LOSS-based attack has the highest ASR across all settings.” → Is this the case? From Table 1, it seems for example for DP-Decoding lambda = 0.1 this is not the case?
10. Page 9: “Increasing m tightens the theoretic privacy (per-group ϵ decreases with m; Thm. 4), but it also increases the effective weight of the public distribution in p_final, i.e., more of the public view leaks through.” → Why does it increase the effective weight of the public distribution in p_final?

---

> ### Author Response · Authors · 2025-11-21
> **We thank the reviewer for their detailed and constructive feedback.**
>
> We appreciate your positive assessment of our formalization, threat model, and DP mechanism, as well as your recognition of the substantial improvements in both privacy and utility that DP-Fusion offers over existing DP methods. We have done our best to address all of your comments, including running the suggested experiments, which have genuinely helped us improve this paper.
>
> ## Responses to weaknesses:
>
> > 1.  ... How exactly is DP Fusion different from the mentioned methods ...
>
> Section 4 contains a high-level comparison between all surveyed methods, which we kept short due to space constraints. We will expand in more detail here and clarify the differences in our revised paper.
>
> PATE is an approach for private model training that uses an ensemble of models, each trained on disjoint subsets of a private dataset, to produce labels under DP guarantees, following the sample-and-aggregate framework. SUBMIX extends PATE to generative tasks by returning samples from a distribution rather than class labels, as done in PATE’s discriminative/classification setting.
>
> A key limitation of SUBMIX is that its ensemble can exceed the privacy budget before completing all $T$ queries, due to data-dependent accounting. PMixED addresses this by adopting Rényi Differential Privacy (RDP) with closed-form privacy accounting.
>
> Our work extends the prior training‑time only, PMixED approach to the LLM inference setting under a fundamentally different threat model and privacy game, where the defender paraphrases a document for privacy and the attacker attempts to recover private tokens.
>
> Our experiments demonstrate a major vulnerability (LOSS Attack, ASR 60%) in the way probabilistic models (LLMs) are used to provide privacy for sensitive data (private documents). While defenses such as differential privacy offer a provable worst-case guarantee, which is a desirable property, our results show that existing DP approaches strongly deteriorate utility to the point where they are essentially unusable. To address this limitation, we introduce a defense under the proposed framework of Differentially Private Inference (DPI), which applies the inference algorithm from PMixED to get theoretical privacy on private tokens in the LLM input. Our proposed mechanism, DP-Fusion, achieves a substantially improved privacy/utility trade-off than the other surveyed methods.
>
>
> > 2. ... LLM has access to the private tokens as well? ...
>
> We thank the reviewer for catching this important oversight. We acknowledge that the notation in the algorithm was confusing and could incorrectly suggest that the full document $D$ (including private tokens) was passed when computing the public distribution. In practice, $D'$ refers only to the **accumulated generated tokens**, not the full document. Specifically, during paraphrasing (see Appendix A.8), the context is built using the paraphrasing query $Q$ and either the original or privacy-group–redacted document, and generated tokens are appended incrementally after the prefix (“Sure, here is … placeholders:”).
>
> We have corrected this by redefining $D'$ in line 3 of the algorithm to denote only the generated tokens ($D_{\text{out}}$) and not include $D$. We have also verified that our implementation already follows this correct behavior, and have revised the paper to correct the oversight.
>
> > 3. ... how does min-k actually infer the secrets once the min k tokens have been identified? ...
>
> We thank the reviewer for this helpful suggestion and will provide more details on the attacks. Both attacks aim to model the probability of observing a given paraphrase conditioned on which secret token set was present in the input. The attacker varies the context by replacing the private tokens with each candidate set from $C_j^*$ and computes the likelihood of generating the observed paraphrase under that context. In the Min-K attack, this likelihood is estimated using the minimum probability among the top-$k$ tokens in the observed paraphrase. In the LOSS attack, it is estimated using the full-sequence perplexity (negative log-likelihood) of the paraphrase. The candidate set yielding the highest probability is selected as the attacker’s prediction for the true secret:
>
> $\arg\max_{C_j^\ast} P(\text{output paraphrase} \mid \text{document with candidate PII token set from } C_j^\ast)$
>
> We will update Section 4.3 to clarify this in more detail.

---

> > ### Author Response · Authors · 2025-11-21
> > **Response to other weaknesses**
> >
> > > 4. ... few more models ...
> >
> > We focused on Qwen2.5-7B because it is widely used, publicly accessible, and, as we observed early in this work, handles tokens sampled from mixed distributions more reliably, which motivated its selection as our primary model. However, following the reviewer’s suggestion, we add experiments to evaluate two additional models from different model families with comparable parameter sizes.
> >
> > We evaluate two additional models from different families with comparable parameter sizes:
> >
> > * "mistralai/Mistral-7B-Instruct-v0.3"
> > * "meta-llama/Meta-Llama-3-8B-Instruct"
> >
> > To evaluate these models, we use the same LLM-as-a-judge setup described in Section 5.3. We report win rate (higher is better) relative to the "Qwen/Qwen2.5-7B-Instruct" model used in the paper. The results are shown below:
> >
> > | αβᵢ  | Mistral-7B | Llama-8B |
> > |-|-|-|
> > | 0.001|39|39|
> > | 0.01 |41|38|
> > | 0.1  |42|43|
> > | 5.0  |20|17|
> > | 10.0 |29|18|
> >
> >
> > Across all settings, we empirically find that Qwen2.5-7B consistently preserves higher utility when used as the base model for DPI, which is also consistent with many benchmarks that shows Qwen2.5-7B outperforming the other two models. The other models perform worse at higher $\alpha\beta_i$ and interestingly, Llama degrades more noticeably compared to Mistral-7B. We cannot make a definitive statement *why* Qwen2.5-7B performs better, but note that it is consistent with many benchmarks that put Qwen2.5-7B ahead of the other models.
> >
> > We will revise the paper to add these experiments and our interpretation of the results to the appendix.
> >
> >
> > > 5. ... standard errors ...
> >
> > We thank the reviewer for this suggestion which will further strengthen the paper. We are in the process of running multiple repetitions of experiments and will upload the revised paper with errorbars. Based on preliminary results from these reruns, we do not expect the main results or conclusions to change, as we observe only small variation across runs and the overall trends remain consistent.
> >
> >
> > > 6. ... does roughly the same as just No DPI - NER ...
> >
> > We agree with the reviewer that, under the chosen dataset and metrics, the gap between DP-Fusion and the No-DPI NER baseline is small (i.e., there is a marginal gain in utility from using DP-Fusion). However, we would like to highlight that this gap is expected to increase as more sensitive tokens (e.g., PII) are contained in the source document, since DP-Fusion can preserve *some* utility from each private token whereas No-DPI NER preserves no utility. We were limited by the availability of datasets, but we ran additional experiments using a single privacy group ($m=1$) and on a different dataset. Our results show how the gap between No-DPI NER and DP-Fusion is a function of the number of sensitive tokens in the source documents.
> >
> >
> > The single-group implementation of DP-Fusion is described in Appendix A.19. We also include an alternative utility metric following prior work [1], using cosine similarity between text embeddings from *sentence-transformers/all-MiniLM-L6-v2*. We compute similarity (higher is better) between each generated paraphrase and the original document.
> >
> > At $\alpha\beta_{i}=0.01$, DP-Fusion achieves higher utility than No DPI–NER (0.816 vs. 0.8093 cosine similarity) while also providing lower LOSS ASR (0.263 vs. 0.277).
> >
> > On the medical MACCROBAT dataset (Appendix A.20) which contain more PII, we observe a much larger gap in utility. DP-Fusion paraphrases are substantially more similar to the original documents (0.6348 vs. 0.4972) while maintaining comparable LOSS ASR (0.125 vs. 0.117). We believe that in this dataset, sensitive tokens are more important to create 'good' paraphrases which explains the larger gap between No-DPI NER (0.4972) and No-DPI Original Document (0.8396), compared to the much smaller gap in ECHR (0.8092 vs. 0.8254).
> >
> > We thank the reviewer for raising this point and will revise the paper to include this discussion.
> >
> > [1]  Lau et al. Understanding the Effects of Human‑written Paraphrases in LLM‑generated Text Detection. 2024.

---

> > > ### Author Response · Authors · 2025-11-21
> > > **Responses to questions**
> > >
> > > > 1. ... Where are the experiments for this ...
> > >
> > > We thank the reviewer for raising this point and apologize for the confusion. The evaluated attacks in Section 4.3 are specifically designed to measure this highlighted vulnerability. These attackers model the probability of observing a given paraphrase conditioned on which secret token set was present in the input, requiring knowledge of the model used for paraphrasing to accurately estimate this probability. Please refer to Table 1 for the summarized attack success rates, with detailed ASR results provided in Appendices A.12, A.13, and A.14.
> > >
> > >
> > > > 2. ... Where are the experiments for this in the paper? ...
> > >
> > > Since DP-Fusion operates entirely at inference time (no model training or finetuning), and the attacker knows the exact same model used for generation, this is closer to a white-box setup (since the attacker has no uncertainty beyond not knowing the tokens they want to infer). Please refer to our previous response for additional details on the attack setup and the corresponding experimental results. We will clarify this in the revised paper.
> > >
> > > > 3. ... symmetric Renyi divergence ...
> > >
> > > We use the **symmetric Rényi divergence**, defined as
> > >
> > > $D_{α}^{↔}(P∥Q) = \max { D_{α}(P∥Q),; D_{α}(Q∥P) },$
> > >
> > > because our privacy notion is based on **add/remove group adjacency**. A neighboring pair of documents ((D, D′)) may differ by either including or excluding a privacy group (X_i).
> > > Differential privacy must hold in both directions (“with–vs–without” and “without–vs–with”), so a single symmetric condition is needed to upper-bound divergence regardless of which document plays the role of (D) or (D′).
> > >
> > > In Algorithm 1 (line 6), we therefore choose the mixing coefficient (λ_i) as the largest value satisfying
> > >
> > > $D_{α}^{↔}\bigl(λ_i p_{\text{priv},i} + (1 - λ_i) p_{\text{pub}} || p_{\text{pub}}\bigr) \le α β_i.$
> > >
> > > This mirrors the **RDP-mollifier ball** construction in Flemings et al. (2024, Definition 4.2; Lemma 4.1): by bounding the symmetric Rényi divergence to the public distribution, we ensure that any two mollified distributions lie within the same RDP “ball.”
> > > In our setting, this corresponds exactly to bounding how much the output distribution can change when the tokens of group (X_i) are added or removed, our group-level neighborhood definition.
> > >
> > > We thank the reviewer for highlighting this and would like to reassure that we have added a formal definition of the symmetric Rényi divergence immediately after Definition 4 and before Theorem 4, along with an explanation of its relevance to our group-level adjacency setting.
> > >
> > >
> > > > 4. ... how \beta is a proxy for \epsilon?
> > >
> > > In our experiments, we fix $\alpha = 2, \delta = 0.001$. For a fixed $\alpha$, and $\delta$, number of privacy groups $m$, the resulting $\epsilon$ for the generated tokens is primarily varied by controlling $\beta$ ( Theorem 4 ). $\alpha\beta$ is the constant that bounds the divergence between the mixed distribution and the public distribution (line 6, algorithm 1). Intuitively, increasing this value allows the mixed distribution to deviate further from the public distribution, i.e., it permits a higher $\lambda$, thereby incorporating more private information. Thus this value acts as the main knob to transition from a more privacy-focused, low-utility setting to a lower-privacy, high-utility setting.
> > >
> > > We will revise Section 4 to include the above discussion on $\beta$ for greater clarity. Thank you for highlighting this.
> > >
> > > > 5. ... Why is \alpha set to 2?
> > >
> > > We observe greater stability when using $\alpha = 2$ with regards to the divergence, which can be effectively controlled with small $\lambda$ values, as shown in Appendix A.22. We therefore adopt this setting for all our experiments. This choice is also consistent with prior work in differential privacy, where ($\alpha = 2$) is commonly used for simplicity. Nevertheless, we conduct an ablation study across different ($\alpha$) values while fixing ($\beta = 0.01$). We evaluate paraphrase quality using our LLM-as-a-judge metric (Section 5.3), reporting comparisons relative to the ($\alpha = 2$) paraphrases as seen below:
> > >
> > > |α|ε|Win-Rate|
> > > |-|-|-|
> > > |1.5|88.87|48|
> > > |2.0|92.02|50|
> > > |2.5|88.74|45|
> > > |3.0|78.69|60|
> > >
> > > Epsilon is similar for most $\alpha$ values but lower at ($\alpha = 3.0$), which also achieves the highest win rate. We will revise the paper to include this ablation in the Appendix.
> > >
> > > > 6. ... upper bound on the advantage ...
> > >
> > > We appreciate the reviewer for making this important observation. Equation 4 is difficult to theoretically upper-bound because it includes baseline leakage, and we do not know which samples the model has previously encountered during training or which tokens can be predicted accurately from a prior. Therefore, we measure the advantage in the token-recovery game empirically.

---

> > > > ### Author Response · Authors · 2025-11-21
> > > > **Responses to other questions**
> > > >
> > > > > 7. ... LLM is specifically asked to ensure privacy?
> > > >
> > > > We would like to clarify that this is only the part of the prompt that is added on top of the existing engineered prompt (described in Appendix A.8) to specifically ask the LLM to ensure privacy. The base prompt was carefully designed to ensure that the redacted text, represented by placeholders (i.e $\_$), does not degrade utility. After testing multiple variations, we converged on this variant as it provided the best utility for the public baseline, especially given that it operates on redacted input where simpler alternatives often produced garbled text.
> > > >
> > > > We will update Section 5.1, the part introducing empirical defenses, for improved clarity.
> > > >
> > > >
> > > > > 8. ... epsilons of different methods are comparable or no? ...
> > > >
> > > > Our paper includes theoretical privacy (ε) vs. utility plots for each method (Figures 6 and 7), which follow the classical comparison method. However, we do not report ε values in the main results table because the DP guarantees across methods are fundamentally different. DP-Fusion provides add-or-remove k-token local DP, whereas methods such as DP-Prompt provide add-or-remove document-level local DP, and we believe directly listing these values without context could be misleading. Furthermore, the theoretical ε for prior DP methods is extremely high (for example, >10,000 for DP-Prompt and >6,400 for DP-Decoding) compared to a maximum of 65 for DP-Fusion, which renders such comparisons uninformative. Therefore, we instead focus on empirical privacy leakage (ASR) as a more meaningful and interpretable measure.
> > > >
> > > > > 9. ... for DP-Decoding lambda = 0.1 this is not the case ...
> > > >
> > > > We thank the reviewer for the observation. The statement refers to the overall mean across all evaluated settings and methods. As shown below, the LOSS-based attack achieves the highest average ASR overall:
> > > >
> > > > | Attack Metric | Overall Mean ASR |
> > > > | ------------- | ---------------: |
> > > > | LOSS          |       **0.3567** |
> > > > | MIN5%         |       **0.2667** |
> > > > | MIN10%        |       **0.2753** |
> > > > | MIN20%        |       **0.3037** |
> > > > | MIN40%        |       **0.3290** |
> > > >
> > > > > 10. ... Why does it increase the effective weight of the public distribution ...
> > > >
> > > > The effect comes directly from the mixture structure in Algorithm 1.
> > > >
> > > > For each group (i), DP-Fusion forms
> > > > $p_i=\lambda_i,p_{\text{priv},i}+(1-\lambda_i),p_{\text{pub}},$
> > > >
> > > > and the final distribution is their average:
> > > > $p_{\text{final}}=\frac{1}{m}\sum_{i=1}^m p_i.$
> > > >
> > > > Expanding this gives:
> > > >
> > > > $p_{final} = p_{pub}
> > > >     + (1/m) * \sum_{i=1..m} [ \lambda_i * (p_{priv,i} - p_{pub}) ].$
> > > >
> > > > This expansion shows that $p_{pub}$ always appears as the baseline, because it is included in every mixed distribution. In contrast, each group’s private information contributes only through the correction term, which is scaled by $1/m$. As $m$ increases, the public component remains unchanged, while each private component becomes increasingly diluted by this $1/m$ factor. As a result, the final mixture moves closer to the public distribution, meaning the effective weight of $p_{pub}$ in $p_{final}$ grows as the number of groups increases. We will revise the paper to include this discussion in the appendix.
> > > >
> > > > *We value your feedback and thank you for it. We hope our response addresses all of your concerns.*

---

> ### Comment · Reviewer_7yFi · 2025-11-25
>
> Thank you for the extensive rebuttal with several additional experiments. I especially appreciated the extensive explanation regarding the comparison with No DPI - NER and the additional experiments there. Most of my questions and concerns have been addressed and hence I maintain my positive score.
>
> However, I did have a few remaining follow-up questions:
> 1. It seems in line 3 of the algorithm box D' is still set to D U D_{out} even though the authors mentioned they updated it. Shouldn't it just be D' = D_{out}?
> 2. In Figure 3, could the authors comment on the very low win rate of the proposed method against No DPI - Ori. Doc.? Is this expected?
> 3. *"Equation 4 is difficult to theoretically upper-bound because it includes baseline leakage, and we do not know which samples the model has previously encountered during training or which tokens can be predicted accurately from a prior. Therefore, we measure the advantage in the token-recovery game empirically."* In this case, maybe the authors can revise the statement *"Section 4.2 gives an upper bound on the attacker’s advantage measured in the token recovery game"* a bit since section 4.2 is all theorems / definitions and so I was expecting that section to contain a theoretical upper bound on the attacker's advantage, which it does not.
> 4. For the new No DPI - NER experiments, why did the authors switch to use cosine similarity instead of perplexity (which was used in the paper) to measure utility?

---

> > ### Author Response · Authors · 2025-11-28
> >
> > We thank the reviewer for taking the time-out to go through the rebuttal and for maintaining their positive rating.
> >
> > > It seems in line 3 of the algorithm box D' is still set to D U D_{out} even though the authors mentioned they updated it. Shouldn't it just be D' = D_{out}?
> >
> > Yes, you are correct, it should just be $D' = D_{out}$ . We are currently working on incorporating all proposed updates and will upload the revised manuscript in the coming days.
> >
> > > In Figure 3, could the authors comment on the very low win rate of the proposed method against No DPI - Ori. Doc.? Is this expected?
> >
> > Yes, at the 0.01 setting the win-rate is 4%, and at 0.1 it is 7%. This is expected. In the No DPI – Original Document setting, we pass the full document directly for paraphrasing, so private information leaks into the output. Our LLM-judge prompt (Appendix A.10) evaluates similarity to the original document; since No DPI – Original Document is closer to the original, leading to a very low win-rate for DP-Fusion. At the same time, the privacy is very poor: the LOSS ASR is 62.67%. In contrast, DP-Fusion has substantially lower ASR (26%–29.33%). Thus, No DPI – Original Document represents the *highest utility but lowest privacy*.
> >
> > > "Equation 4 is difficult to theoretically upper-bound ...
> >
> > We are revising the draft and will update the paper to clearly reflect that we only measure the attacker’s advantage in the token-recovery game *empirically*. We propose to change the sentence “*Section 4.2 gives an upper bound on the attacker’s advantage measured in the token recovery game*”
> > to “*Section 4.2 gives an empirical estimate of the attacker’s advantage measured in the token recovery game.*”
> >
> > > For the new No DPI - NER experiments, why did the authors switch to use cosine similarity instead of perplexity (which was used in the paper) to measure utility?
> >
> > We include cosine similarity only in the appendix because we wanted to report an additional utility metric beyond the main-paper metric. The main paper continues to use perplexity, which is the standard in prior work. Cosine similarity is simply cheaper to compute, it operates directly on the generated text with almost no extra cost. In contrast, perplexity under ground-truth teacher forcing requires a full extra forward pass separate from paraphrase generation, which is expensive for these additional appendix experiments.
> >
> > We thank the reviewer again for their positive score and constructive discussion that has genuinely helped us improve the work.

---

### Official Review · Reviewer_EZSD · 2025-11-01

**Soundness:** 3
**Presentation:** 3
**Contribution:** 3
**Rating:** 6
**Confidence:** 4

**Summary:**

This paper introduces DP-Fusion, a differential privacy (DP) mechanism to protect user privacy during large language models (LLMs) inference. The approach bounds the influence of sensitive tokens on the generated output to protect sensitive information from attackers. Empirical results show that their method has substantially better privacy-utility trade-offs compared to prior differential privacy inference baselines, such as DP-Prompt and DP-Decoding.

**Strengths:**

- The authors propose a clever way to bound the influence of sensitive tokens during document paraphrasing with formal privacy guarantee.
- The empirical evaluations are thorough, and DP-Fusion has demonstrated substantial gains over standard DP methods.
- The paper is clear and well-written.

**Weaknesses:**

- Besides perplexity, the authors are suggested to measure the performance on downstream task to show the paraphrase utility.
- DP-Fusion requires the user to paraphrase the document for $m+1$ times, and the user-side complexity should be thoroughly discussed.

**Questions:**

- In Table 1, why DP-Fusion offers better privacy protection than No DPI - NER (in terms of ASR), which directly remove the sensitive information?

---

> ### Author Response · Authors · 2025-11-20
> **We thank the reviewer for their valuable feedback.**
>
> The reviewer likes that DP-Fusion delivers 'substantially better' privacy/utility trade-offs over existing methods and that our method is 'clever' and 'thoroughly evaluated'. They also raise that our paper is 'well written', which we appreciate. We address both weaknesses raised by the reviewer below by running additional experiments and will revise the paper to address them.
>
> > ... measure the performance on downstream task ...
>
> We thank the reviewer for raising this point. We agree that evaluating the downstream performance of the paraphrased documents would strengthen our paper. We did not do so initially due to a lack of suitable datasets for such an assessment, which is why we focused on (1) perplexity (PPL), (2) Win Rates with LLM Judges and (3) qualitative assessments.
>
> To address the reviewer's suggestion, we created a custom  a multiple-choice questionnaire on ECHR (described in Section 5.1 and Appendix A.5) to evaluate downstream performance. We sample 200-token (Qwen-2.5 tokenizer) chunks from each document with the following statistics:
>
> | Metric            | Mean  | Std  | Min  | Max  |
> |-------------------|-------|------|------|------|
> | Lines/chunk       | 5.09  | 2.17 | 1    | 12   |
> | Tokens/chunk      | 200   | 0    | 200  | 200  |
> | Private toks/chk  | 27.80 | 5.83 | 20   | 47   |
> | Private %         | 13.90 | 2.91 | 10.0 | 23.5 |
> | Entities/chunk    | 10.77 | 2.69 | 6    | 21   |
>
>   Entity Type Distribution (1,077 total entities across 100 chunks):
>
> | Entity | Cnt | %     |
> |--------|-----|-------|
> | DATE   | 499 | 46.33 |
> | PERSON | 191 | 17.73 |
> | ORG    | 173 | 16.06 |
> | LOC    | 87  | 8.08  |
> | QTY    | 52  | 4.83  |
> | DEM    | 38  | 3.53  |
> | MISC   | 30  | 2.79  |
> | CODE   | 7   | 0.65  |
>
>
> We then define questions for each chunk of the form, by prompting OpenAI's GPT-4o:
>
>     "Which specific detail is explicitly supported by the excerpt?",
>     "Which identifying fact appears verbatim in the passage?",
>     "Which of the following details can be confirmed from the excerpt?",
>     "Which factual statement matches the information given in the passage?",
>     "Which claim is directly grounded in the excerpt?"
>
> Again, we use GPT-4o to generate the correct option and the distractors.
>
> We then use different DP methods and baselines to generate a privatised version of each chunk and evaluate them in the following chat template:
>
> ```
>     "<|im_start|>system
>     Select the correct option based on the passage provided below. You must output one token i.e A,B,C,D that's it nothing else. Do not output any new lines.
>     {system_prompt}<|im_end|>
>     <|im_start|>user
>     Passage: {passage}
>     Question: {question}
>     Options: A) {options[0]}, B) {options[1]}, C) {options[2]}, D) {options[3]}<|im_end|>
>     <|im_start|>assistant
>     The answer token is:"
> ```
>
> We measure accuracy by extracting the option selected by the LLM (i.e., the token appearing after “The answer token is:”) and compare it with the correct answer.
>
> The below table shows the (i) accuracy and (ii) ASR with the LOSS attack (Section 4.3) with different privatization methods surveyed in the paper. High accuracy and low ASR are preferable.
>
> | Method      | Parameters               | Accuracy | ASR % - LOSS   |
> | ----------- | ------------------------ | -------- | ------ |
> | No DPI      | Original Document        | 98%      | 62.70% |
> | No DPI      | NER                      | 34%      | 27.70% |
> | DP-Decoding | λ = 0.1                  | 23%      | 15.67% |
> | DP-Decoding | λ = 0.9                  | 70%      | 66.00% |
> | DP-Prompt   | w = 5, T = 0.75          | 31%      | 26.67% |
> | DP-Prompt   | w = 5, T = 1.75          | 32%      | 17.33% |
> | DP-Prompt   | w = 50, T = 0.75         | 90%      | 56.67% |
> | DP-Prompt   | w = 50, T = 1.75         | 33%      | 28.67% |
> | DP-Fusion   | $\alpha\beta_{i}$ = 0.001| 36%      | 28.00% |
> | DP-Fusion   | $\alpha\beta_{i}$ = 0.01 | 37%      | 26.00% |
> | DP-Fusion   | $\alpha\beta_{i}$ = 0.1  | 38%      | 29.30% |
> | DP-Fusion   | $\alpha\beta_{i}$ = 5.0  | 60%      | 45.70% |
> | DP-Fusion   | $\alpha\beta_{i}$ = 10.0 | 85%      | 56.30% |
>
> For DP-Fusion, we use the faster single-group setting described in Appendix A.19. We observe that DP-Fusion achieves the best privacy/utility trade-off. At ($\alpha\beta_i$ = 0.01), DP-Fusion offers better trade-offs than No DPI–NER, achieving higher utility (34% vs. 37%) at similar empirical privacy levels (lower ASR 27.7% vs. 26.3%). As $\alpha\beta_i$ increases, the privacy/utility trade-offs interpolates between the No DPI–NER setting ($\alpha\beta_i=0$) toward the No DPI–Original Document ($\alpha\beta_i=\infty$) setting.
>
> We will revise the paper to include these experiments  in the appendix and release the dataset. We will also add the multiple-choice evaluation prompt and the corresponding results.

---

> > ### Author Response · Authors · 2025-11-20
> > **Response to other weaknesses and questions.**
> >
> > > DP-Fusion requires the user to paraphrase the document for $m+1$ times ...
> >
> > The reviewer is correct in that our private inference method requires $m+1$ forward passes per token (where $m$ is the number of privacy groups) as opposed to $1$ forward pass in the non-private case. We would like to note that our method is highly parallelizable and the latency is approximately equivalent to that of a single LLM forward pass. We also performed experiments in the single-group setting ($m=1$) in Appendix A.19 of our paper, which requires fewer computational resources and mollifies only between the public (document minus private tokens) and private (original document) contexts.
> >
> > We will revise the paper to include a discussion about the increased computational load required for our method.
> >
> >
> > > ... why DP-Fusion offers better privacy protection than No DPI - NER ...
> >
> > We thank the reviewer for raising this point. We believe the slightly better privacy performance of DP-Fusion compared to No-DPI NER arises from the additional randomness introduced during distribution mixing (Algorithm 1), which adds noise and marginally reduces ASR. This effect, lower ASR at the highest-privacy setting ($\alpha\beta_i$ = 0.01), also appears in the single-group implementation (Appendix A.19) and on a different dataset (Appendix A.20). However, the difference remains small in all cases.
> >
> > *We value your feedback and thank you for it. We hope our response addresses all of your concerns.*

---

> > > ### Comment · Reviewer_EZSD · 2025-11-23
> > >
> > > Thanks for the rebuttal. The authors have addressed my concerns and I raised the scores accordingly.

---

> > > > ### Author Response · Authors · 2025-11-23
> > > >
> > > > We thank the reviewer for taking the time to read our rebuttal and for updating their scores. Please feel free to comment if you have any further questions.

---

### Comment · Area_Chair_XhPE · 2025-11-25

Dear Reviewers,

The authors have submitted their responses to your questions and feedbacks. Please read them and give your comments.

Regards, AC

---

### Author Response · Authors · 2025-12-03
**Summary of Discussion and Updates**

In this paper, we present a method to privately infer an LLM while bounding the influence of any sensitive token using differential privacy (DP). Our method works by mixing private and public distributions, and we substantially outperform all other surveyed DP methods.

We thank the reviewers for their detailed feedback that has allowed us to substantially improve this work. We have uploaded a revised draft with the changes (marked in blue) based on the reviewer’s comments.

> Performance on a downstream task

We have added Appendices A.23 and A.24 explaining the creation of a new multiple-choice questionnaire using ECHR documents (Section 5.1) with PII in context, which we use to evaluate DP-Fusion both as a private-context sanitisation method and as a privatisation mechanism in a live-chat setting.

> Clarification on compute/run-time.

We have added a brief discussion at the end of Section 4.1 to address this point.

> Comparison with baselines

We have added an explanation in Section 5.4 on why DP-Fusion can offer better privacy protection than No DPI–NER at certain settings, as well as a discussion in Section 6 analysing the performance gap when comparing against the No DPI–NER baseline.

> Differences with existing methods

We have updated Section 4 to include a more detailed overview of existing methods and how DP-Fusion compares to them.

> Evaluating different models

We have included these experiments in Appendix A.25.

> Clarifications on theoretical privacy

We included Definition 5 on symmetric Rényi divergence, added clarification of the DP parameters in Section 4, and included an ablation over α in Appendix A.26.

> The role of tagging

We have updated Section 6 to explain this in more detail.

> Clarifications on prompt and placeholders

We have updated Section 5.1 to further explain the prompt for empirical defenses and added a clarification on placeholder tokens in Section 4.1.

> Fixing minor errors/typos.

We have fixed the variable-assignment issue in Algorithm 1 and corrected the citation issues.

*We thank the reviewers again, as well as the AC, for going through the detailed discussion.*

---

### Meta-Review · Area_Chair_xWzH · 2025-12-06

**Summary:**

This paper introduces DP-Fusion, a differential privacy (DP) mechanism to protect user privacy during large language models (LLMs) inference. It addresses the limitations of previous works: (i) lack provable guarantees or (ii) have a poor utility/privacy trade-off. DP-Fusion works as follows: (1) label a subset of sensitive tokens, (2) infer the LLM without any sensitive tokens to obtain a baseline, (3) infer the LLM with the sensitive tokens, and (4) blend distributions so that the final output remains within a bounded distance of the baseline distribution.
Experimental results show that the proposed method creates token-level provably privatized documents with substantially improved theoretical and empirical privacy, achieving lower perplexity than related DPI methods.

At the initial rating, three reviewers are positive while one reviewer is negative. The reviewers raise several concerns, including: Performance on a downstream task, Clarification on compute/run-time, Comparison with baselines, and so on. The authors provide a sufficient response that addresses most of the concerns, especially the concerns of the negative reviewer.

Therefore, the AC recommends Accept.

**Reviewer Concerns:**

The authors have addressed most of the concerns:
1. Performance on a downstream task.
2. Clarification on compute/run-time.
3. Comparison with baselines
4. Differences with existing methods
5. Evaluating different models
6. Clarifications on theoretical privacy
7. The role of tagging
8. Clarifications on prompt and placeholders
9. Fixing minor errors/typos.

Although concerns about the scope and experiment tasks are raised, the AC thinks the authors provide a reasonable response. The authors should add more related discussion in the revision.

**Reviewer Scores:**

The AC thinks the final discussion may be overall positive.

---

### Decision · Program_Chairs · 2026-01-26

Accept (Poster)